

# Migration and transformation of dissolved carbon during accumulated cyanobacteria decomposition in shallow eutrophic lakes: a simulated microcosm study

Zhichun Li[1,2,4], Yanping Zhao[3], Xiaoguang Xu[3], Ruiming Han[3], Mingyue Wang[3] and Guoxiang Wang[1,3]

[1] School of Geography Science, Nanjing Normal University, Nanjing, Jiangsu Province, China
[2] School of Environment and Surveying Engineering, Suzhou University, Suzhou, Anhui Province, China
[3] School of Environment, Nanjing Normal University, Nanjing, Jiangsu Province, China
[4] National Engineering Research Center of Coal Mine Water Hazard Controlling, Suzhou, Anhui Province, China

Corresponding authors
Xiaoguang Xu, xgxu@njnu.edu.cn
Guoxiang Wang,
wangguoxiang@njnu.edu.cn

## ABSTRACT

The decomposition processes of accumulated cyanobacteria can release large amounts of organic carbon and affect the carbon cycling in shallow eutrophic lakes. However, the migration and transformation mechanisms of dissolved carbon (DC) require further study and discussion. In this study, a 73-day laboratory microcosm experiment using suction samplers (Rhizon and syringe) was conducted to understand the migration and transformation of DC during the cyanobacteria decomposition. The decomposition of cyanobacteria biomass caused anoxic and reduction conditions, and changed the acid-base environment in the water column. During the early incubation (days 0–18), a large amount of cyanobacteria-derived particulate organic matter (POM) was decomposed into dissolved organic carbon (DOC) in the overlying water, reaching the highest peak value of 1.82 g L$^{-1}$ in the treatment added the high cyanobacteria biomass (470 g). After 18 days of incubation, the mineralization of increased DOC to dissolved inorganic carbon (DIC) maintained a high DIC level of overlying water in treatments added cyanobacteria biomass. The treatment added the medium cyanobacteria biomass (235 g) presented the lower DOC/total dissolved carbon ratio than the high cyanobacteria biomass associated with the lower mineralization from DOC to DIC. Due to the concentration differences of DIC at water-sediment interface, the main migration of DIC from pore water to overlying water occurred in the treatment without added cyanobacteria biomass. However, the treatments added the cyanobacteria biomass presented the obvious diffusion of DOC and the low migration of DIC at the water-sediment interface. The diffusive fluxes of DOC at the water-sediment interface increased with the cyanobacteria biomass added, reaching the maximum value of 411.01 mg/(m$^2$·d) in the treatment added the high cyanobacteria biomass. In the overlying water, the group added the sediment and medium cyanobacteria biomass presented a faster degradation of cyanobacteria-derived POM to DOC and a higher mineralization level of DOC to DIC than added the medium cyanobacteria biomass without sediment. Therefore, during accumulated cyanobacteria decomposition, the biomass of accumulated

cyanobacteria and sediment property can influence the migration and transformation of DC, playing an important role in carbon cycling in shallow eutrophic lakes.

## INTRODUCTION

Carbon cycling in lakes, from a macro perspective, refers to the constant migration and transformation between gaseous carbon, dissolved carbon (DC) and solid or particulate carbon (*Kuntz et al., 2015*; *Quay et al., 1986*). DC can be found in the whole water column, is an important media for the transformation of other carbon forms and is conducive to absorption and utilization by aquatic plants and microorganisms (*Bass et al., 2010*; *Koehler, Broman & Tranvik, 2016*). DC, including dissolved inorganic carbon (DIC) and dissolved organic carbon (DOC), originates from autochthonous production or allochthonous input. DIC ($CO_2$, $CO_3^{2-}$, and $HCO_3^-$) is an important aquatic nutrient that influences the carbon cycle and is mainly produced in sediment pore water during organic matter oxidation. It is also influenced by soil $CO_2$ from catchments, the influx or efflux of $CO_2$ from or to the atmosphere, and the balance between photosynthetic $CO_2$ uptake and respiratory $CO_2$ production (*Vreca, 2003*; *McDonald et al., 2013*; *Bass et al., 2010*). DOC composition has been studied through the quantification of specific compounds, including short-chain organic acids, amino acids, and major classes of biomolecules (*Komada et al., 2013*; *Gómez-Consarnau et al., 2012*; *Bertilsson & Jones, 2003*). In lake ecosystems, the autochthonous and allochthonous DOC refers to various autotrophic and heterotrophic in situ activities and the hydrodynamic processes of catchments, respectively. Allochthonous DOC, typically deriving from vegetation and soil organic matter, is more stable and resistant to degradation than autochthonous DOC (*Catalán et al., 2013*; *Tranvik, 1992*; *Jaffe et al., 2008*). Autochthonous DOC is derived from planktonic production, photosynthesis of macrophytes, and micro-biological degradation of organic matter (*Hu et al., 2011*; *De Almeida Assuncao et al., 2016*). In eutrophic lakes, autochthonous DOC mainly originates from phytoplankton biota detritus, particularly cyanobacteria blooms. The transformation between DOC and DIC includes various biogeochemical processes, for example, photochemical reactions, metal complexation, microbial growth, and nutrient and contaminant transport, which affect the environmental behavior of pollutions and the carbon cycle of shallow eutrophic lakes (*Attermeyer et al., 2014*; *Aarnos, Ylostalo & Vahatalo, 2012*; *Marie et al., 2015*; *Yang et al., 2016a*). Therefore, it is crucial to understand the migration and transformation of DC during cyanobacteria decomposition in shallow eutrophic lakes.

Cyanobacterial blooms frequently occur in shallow eutrophic lakes. Depending on the meteorological and hydrological conditions, a large amount of cyanobacteria biomass may drift and accumulate along the lake shoreline (*Kong & Gao, 2005*; *Yang et al., 2016b*),

resulting in the formation of an active carbon pool. Subsequently, these accumulated cyanobacteria are subjected to sedimentation and decomposition processes, coupled with the migration and transformation of DC. Compared with aquatic plants, cyanobacteria are relatively easily decomposed due to their smaller grain size and more favorable ratio of nitrogen to phosphorus for microbial utilization (*Li, Guan & Liu, 2011*; *Sun, 2013*; *Wang & Chen, 2008*; *Liu et al., 2010*). In addition, the growth period of cyanobacteria is generally shorter than that of aquatic plants, and their growth rates are faster. Therefore, the accumulation, sedimentation, and decomposition of cyanobacteria may accelerate contaminant transport, material cycling (carbon, nitrogen, and phosphorus) and energy flow in lake ecosystems. Studies have previously been conducted on cyanobacteria decomposition related to variability in total organic carbon (*Hanamachi, Hama & Yanai, 2008*; *Ye et al., 2011*), bacterial community dynamics and biodegradability of cyanobacteria-derived OM (*Lee et al., 2016*; *Shi et al., 2017*), qualitative and quantitative variability in specific organic carbon compounds (*Liu et al., 2016*), and the accumulation and degradation of chromophoric dissolved organic matter (*Hulatt et al., 2009*). These studies revealed the contribution of cyanobacterial decomposition to the carbon cycle and associated influence factors, including microbial community characteristics, sunlight, temperature, and aerobic and anaerobic conditions. However, few studies have attempted to systemically explore the mechanisms of migration and transformation of DC and to calculate a balance between the aquatic phase and the sediment phase during cyanobacteria decomposition.

This study seeks to describe DC release characteristics in overlying water, and the migration and transformation processes during cyanobacteria decomposition in shallow eutrophic lakes. Samples of cyanobacteria and sediment cores were collected from Taihu Lake for laboratory incubation. Variability in DC concentration from the overlying water, bottom water, and pore-water was monitored depending on the thicknesses of the accumulated cyanobacteria layer (cyanobacteria biomass) and the absence or presence of sediments. DOC diffusive flux at the sediment-water interface was calculated by Fick's first law. It was hypothesized that the biomass of accumulated cyanobacteria and the physical, chemical, and biological structures of sediment significantly influence the DOC release intensity, the transformation between DOC and DIC, and DC diffusion processes at the water-sediment interface in shallow eutrophic lakes. These observations may be beneficial to understand the fates of accumulated cyanobacteria and the carbon cycle of shallow eutrophic lakes.

## MATERIALS AND METHODS

### Sample collection

As the third largest freshwater lake in China, Taihu Lake is a typical shallow and eutrophic lake where cyanobacteria blooms frequently occur in summer. The summer monsoon drives cyanobacteria blooms to drift and accumulate to lakeshore, forming a high biomass of cyanobacteria scum in aquatic macrophyte-belts (Fig. 1). The accumulated cyanobacteria rarely spreads more widely across the sediment because of the obstruction by aquatic macrophyte-belts and their forming trenches. Subsequently, the massive

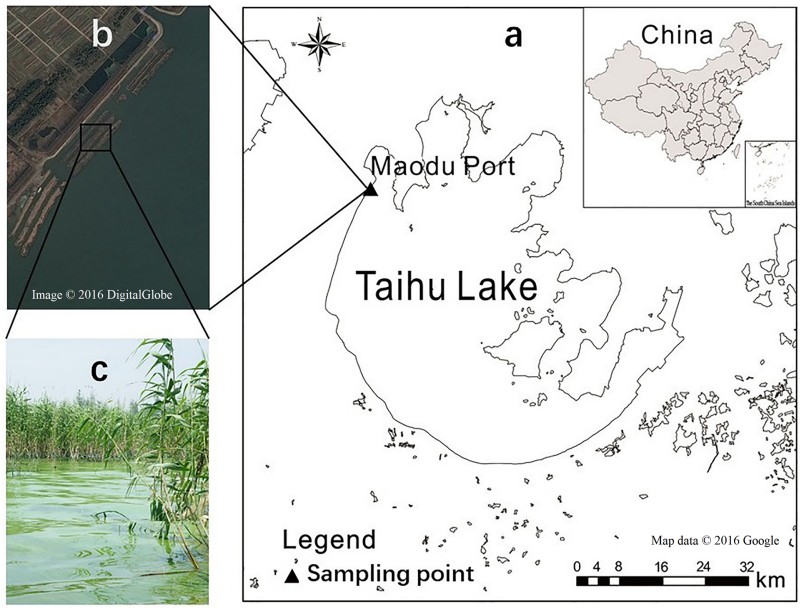

**Figure 1 Location of the sampling sites, Google Maps of trenches, and accumulated cyanobacteria in aquatic macrophyte-belts in the western shoreline of Taihu Lake, China.** Location of the sampling sites (A), Google Maps of trenches (B), and macrophyte-belts (C). Map credit: © 2016 Google; Digital Globe.

decline and decomposition of cyanobacteria blooms in high temperatures probably cause negative impacts on the aquatic ecosystem, including the malodorous water, the death of fish and other aquatic organisms and even the occurrence of black bloom (*Smith, Boyer & Zimba, 2008*; *Feng et al., 2014*; *Liu et al., 2017*).

The sampling operation was executed near the Maodu River (N31°24′42.21″, E120°00′38.92″) along the western shoreline of Taihu Lake in July, 2016 (Fig. 1), including open water and accumulated cyanobacteria area. The overlying water samples from two areas were collected after measuring physical and chemical parameters by calibrated probes from American Hach. The initial physical and chemical parameters as well as the initial concentrations of total nitrogen (TN), total phosphorus (TP), ammonia nitrogen ($NH_4^+$-N), nitrate nitrogen ($NO_3^-$-N) and DOC of the lake water in the two areas were showed in Table 1. The fresh cyanobacteria were sampled by using a 64-μm plankton net. Three intact core sediments were collected using a KC Kajak core sampler equipped with a plexi-glass tube (inner diameter, 8.8 cm). The overlying water and particulate matter were slowly sucked up using a small pump into the upper part of plexi-glass tubes that kept 25 cm of sediment below the sediment-water interface. The samples were stored in the refrigerator at 4 °Cn then taken to the laboratory within 8 h.

## Microcosm system

Before the incubation of intact core sediments in the laboratory, the same volume (40 cm × π × $4.4^2$ cm$^2$) of lake water from open water was slowly added to each plexi-glass tube (inside radius, 4.4 cm) with minimal disturbance (Fig. 2). A plexi-glass tube without sediment and Rhizon samplers (*Seeberg-Elverfeldt et al., 2005*; *Shotbolt, 2010*)

**Table 1 Physical and chemical properties and the concentrations of TN, TP, $NH_4^+$-N, $NO_3^-$-N, and DOC in the lake water from accumulated cyanobacteria areas of lakeshore and open water of Taihu Lake.**

| Lake region | Cyanobacterial thickness (cm) | Depth of water (m) | Temperature (°C) | pH | DO (mg $L^{-1}$) | Eh (mv) | TN (mg $L^{-1}$) | TP (mg $L^{-1}$) | $NH_4^+$-N (mg $L^{-1}$) | $NO_3^-$-N (mg $L^{-1}$) | DOC (mg $L^{-1}$) |
|---|---|---|---|---|---|---|---|---|---|---|---|
| Accumulated cyanobacteria areas of lakeshore | 4.0–10.0 | 0.3–0.6 | 27.6–28.4 | 6.7–7.4 | 0.5–1.5 | −222.9 to −64.1 | 4.45–8.89 | 0.49–1.89 | 0.70–5.97 | 0.01–0.56 | 23.08–103.60 |
| Open water | 0–0.2 | 0.7–3.1 | 27.8–28.6 | 6.9–9.3 | 6.1–16.4 | 86.7–219.3 | 1.59–4.51 | 0.24–0.44 | 0.05–0.17 | 0.63–1.64 | 15.09–21.34 |

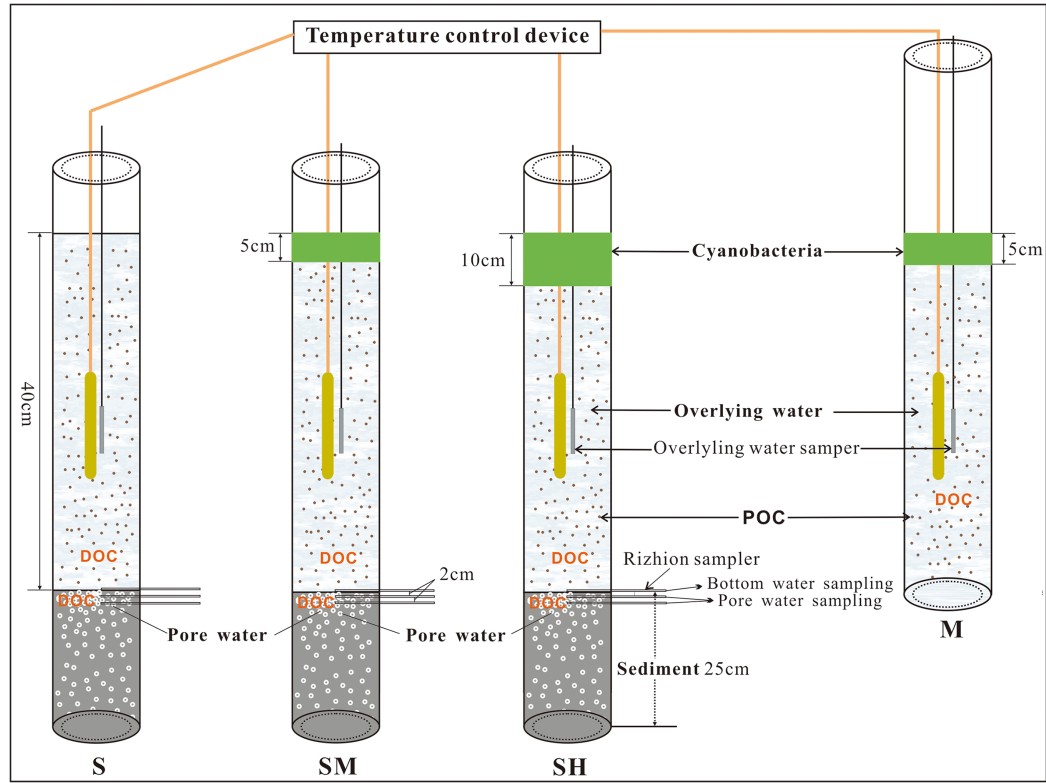

**Figure 2 Microcosm system in in the laboratory for simulating the decomposition of accumulated cyanobacteria in lakeshore of Taihu Lake.** A total of 0, 235, and 470 g cyanobacteria biomass were added into three water columns with sediment named as S, SM, and SH, respectively, while 235 g cyanobacteria biomass was added in the control system without sediment, named M. In the system, 235 and 470 g cyanobacteria biomass can cause five and 10 cm cyanobacteria accumulation layers, respectively.

served as the control. After one week of incubation, 0 g (0 cm), 235 g (5 cm), and 470 g (10 cm) of wet cyanobacteria biomass that had been washed with high purity water were added into three water columns labeled S, SM, and SH, respectively. Meanwhile 235 g (5 cm) of cyanobacteria biomass was added to the control system, labeled M. Cyanobacteria biomass was added based on field observations along the lakeshore where cyanobacteria had accumulated at a thickness of 5–10 cm. The incubation experiment was performed in an air-conditioned lab and was always kept at 28 °C using an automatic constant temperature heating rod. To better represent in-situ conditions in Taihu Lake, the upper parts of the plexi-glass tubes were opened and their entire sides protected from light. To collect pore water, three plexi-glass tubes with 25 cm intact core sediments were fitted to the Rhizon samplers at zero, two, and four cm depths below the sediment-water interface (*Seeberg-Elverfeldt et al., 2005*; *Shotbolt, 2010*).

## Sample analysis

The thicknesses of cyanobacterial layer on surface water and sedimentary detritus layer (cyanobacteria-derived particulates) on surface sediment were measured by straight edge. Dissolved oxygen (DO), pH, and oxidation-reduction potential (Eh) of overlying water

were detected using calibrated probes from American Hach. Syringes were used to extract five mL of overlying water, five mL of bottom water, and 2.5 mL of pore water for the respective groups S, SM, and SH, while five mL of overlying water for group M was collected. Subsequently, the same volume of lake water with extracted water was supplemented into each column. The overlying water samples were filtered through precombusted (470 °C for 4 h) GF/F glass fiber filters (nominal pore size, 0.7 μm; Whatman International Ltd., Maidstone, England). DIC and total dissolved carbon (TDC) values from filtered overlying water and pore water were directly detected using carbon-nitrogen analyzer (multi N/C® 3100; AnalytikJena, Jena, Germany). The carbon–nitrogen analyzer applied the method of combustion oxidation-non-dispersive infrared absorption and the calculation model of minusing, which meant first determining TDC and DIC concentrations and then calculating the DOC concentrations (DOC = TDC–DIC).

### Diffusive flux of DOC at the sediment-water interface

Dissolved organic carbon diffusive flux at the sediment-water interface was calculated according to Fick's first law of diffusion:

$$J = -\phi D_s \frac{\partial C}{\partial x}$$

Where, $\phi$ is the porosity of surficial sediments. According to *Cermelj, Bertuzzi & Faganeli (1997)*, the $\phi$ value in this study was set to 0.68 based on the weight fraction of water after drying the sediments for 24 h at 105 °C. $D_s$ is the bulk sediment diffusion coefficient for DOC, and the Stokes–Einstein equation yielded a value of $1.22 \times 10^{-6}$ cm$^{-2}$ s$^{-1}$ according to the average molecular weight for DOC of 8000 daltons (*Alperin, Albert & Martens, 1994*; *Holcombe, Keil & Devol, 2001*). Therefore, based on the literature on the average molecular weight of pore water DOC (*Burdige et al., 1999*), the $D_s$ was assumed to be $1.22 \times 10^{-6}$ cm$^{-2}$ s$^{-1}$ to assess the DOC diffusive flux in this study. $\partial C/\partial x$ is the DOC concentration gradient between pore-water and bottom water over the applied distance, where $\partial C/\partial x$ can be approximated by $\Delta C/\Delta x$ (*Burdige et al., 1999*; *Lahajnar et al., 2005*). $\Delta C$ is the difference in DOC concentration between bottom water and pore water sampled at two cm depth. $\Delta X$ is the vertical dimension between two sample points (i.e., $\Delta x = 2.0$ cm in the microcosm system).

### Statistical analysis

Temporal variability and box-plots for TDC, DOC, and DIC concentrations and DOC diffusive flux were analyzed using Origin version 8.0 (OriginLab Corporation, Massachusetts, USA) and SPSS statistical package 19.0 (SPSS Inc., Chicago, IL, USA). The correlation analysis between the physicochemical indexes and DC concentrations was conducted using the SPSS statistical package for Windows. The criteria of $p < 0.05$ was considered as the significant level in this study.

## RESULTS AND DISCUSSION

### Visual changes and physicochemical environment associated with DC release

During cyanobacteria decomposition, the thickness of the cyanobacteria layer on superficial overlying water and the thickness of sedimentary detritus layer from cyanobacteria-derived particulates at the bottom of overlying water are shown in Fig. 3. In the treatment without cyanobacteria biomass (group S), the water column remained relatively colorless and transparent during the experiment. However, the overlying water in groups with added cyanobacteria biomass (groups SM, SH, and M) gradually showed variations in visible color and thickness with a foul smell. From day 2 to day 10 there was a slight odor and a large amount of cyanobacteria-derived particulates dispersed into the water column for groups SM, SH, and M. Due to the abundant chlorophyll-a in fresh cyanobacteria (*Shoaf & Lium, 1976*; *Zarel, Sinetova & Cerveny, 2015*), the overlying water displayed the green color, indicating cyanobacteria hadn't been completely decomposed during early incubation. After 10 days, the water color became yellow–green with a strong odor when the cyanobacteria layers in overlying water became incompact and a low aggregation degree. This indicated the massive decomposition of the cyanobacteria-derived particulate organic matter (POM). Subsequently, a number of black particles appeared in the overlying water and sediments, which was similar to previous reports (*Steinberg, 2003*; *Duan et al., 2014*; *Han et al., 2015*) and probably related to the settling of POM. After the 20th day, the sedimentary detritus layer at the bottom decreased in thickness and black particles gradually disappeared. These changes might have occurred with the microbial mineralization of cyanobacteria-derived POM (*Shi et al., 2017*). Therefore, in the treatments added cyanobacteria biomass, the clear color changes in the water column and the morphological changes in the cyanobacteria layer can indicate the degradation level of cyanobacteria-derived POM to some extent.

Variations in DO, Eh, and pH values in the water column according to incubation time are shown in Fig. 4. Based on the initial DO and Eh values, there was no obvious reduction environment in the overlying water, similar to the data of open water in Taihu Lake (Table 1). After adding cyanobacteria, DO concentrations of the water column in groups SM, SH, and M began to rapidly decrease and remained at a lower level (<1.0 mg L$^{-1}$), compared to group S without cyanobacteria biomass. The change trends between DO and Eh were basically consistent, corresponding to the data of accumulated cyanobacteria in Taihu Lake (Table 1). Meanwhile, DOC and DO in groups M, SM, and SH with added cyanobacteria biomass were negatively correlated ($r = -0.328 - -0.310$, $p < 0.05$) (Table 2). This indicated that the DOC release from microbial degradation of organic matter probably consumed oxygen and resulted in anoxic conditions, consistent with previous studies (*Shao et al., 2013*; *Song, He & Jiang, 2016*; *Yan et al., 2017*). The water column was alkaline at the beginning of the experiment. After adding cyanobacteria, the pH values of the water column in groups SM, SH, and M had clear acid-base changes (Fig. 4C). During the first 18 days, the pH values in groups SM, SH, and M were lower and

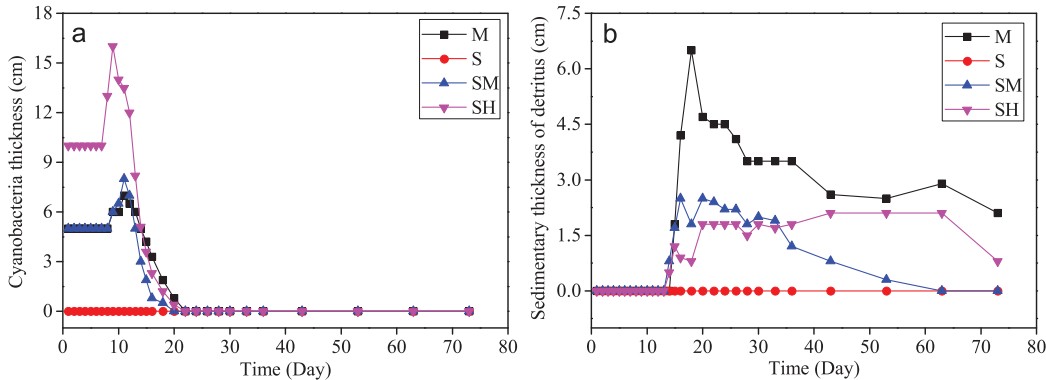

**Figure 3 Variations of cyanobacteria thicknesses of superficial overlying water and the thicknesses of sedimentary detritus layer from cyanobacteria particulates at the bottom of overlying water.** Cyanobacteria thicknesses of superficial overlying water (A). The thicknesses of sedimentary detritus layer from cyanobacteria particulates at the bottom of overlying water (B).

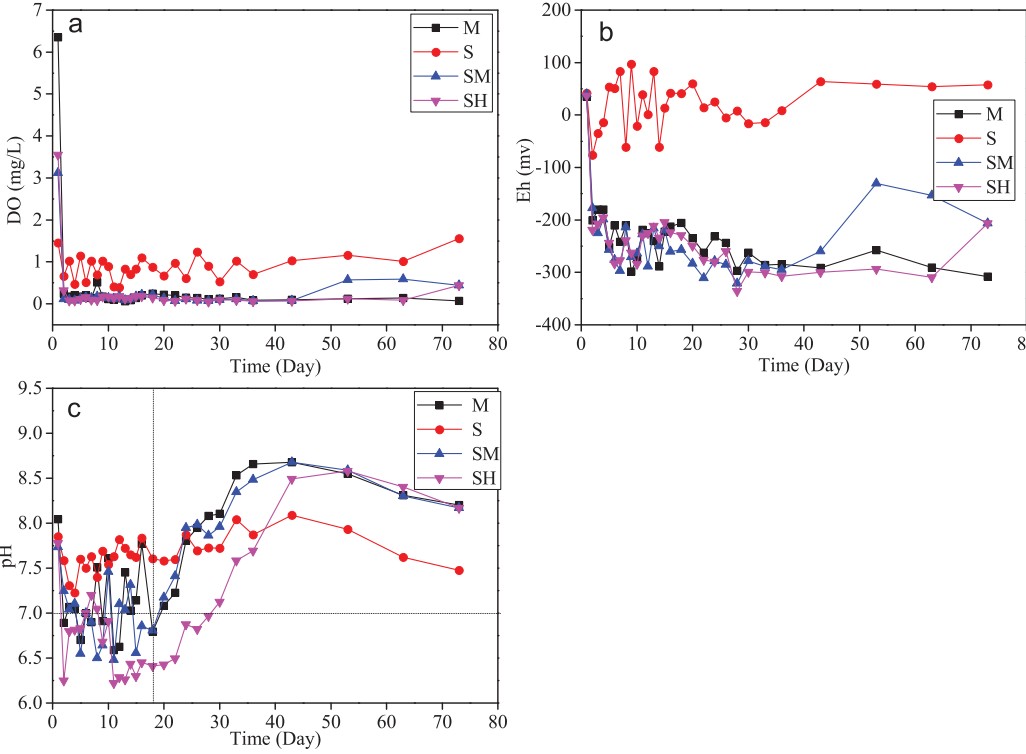

**Figure 4 Variations of DO, Eh, and pH along with the incubation time in each treatment.** DO (A), Eh (B), and pH (C).

fluctuated. Subsequently, they gradually rose to peak values (>8.5) and then dropped after that. In contrast, pH values in group S without biomass fluctuated more than 7.0. DOC and pH in groups SM, SH, and M were negatively correlated ($r = -0.533 - -0.202$, $p < 0.05$), as shown in Table 2. It is well documented that the decomposition of cyanobacteria causes biological, physical, and chemical degradation of proteins, lipids, and

**Table 2 Pearson correlation coefficients between DCs concentrations and physicochemical indexes in different treatments ($p < 0.05$).**

| Treatments | DOC | | | DIC | | |
|---|---|---|---|---|---|---|
| | DOC–DO | DOC–Eh | DOC–pH | DIC–DO | DIC–Eh | DIC–pH |
| S | 0.127 | −0.017 | 0.300 | 0.024 | 0.015 | 0.291 |
| SM | **−0.323** | **−0.407** | **−0.471** | −0.152 | −0.094 | **0.817** |
| SH | **−0.310** | −0.214 | **−0.533** | −0.191 | **−0.423** | **0.581** |
| M | **−0.328** | −0.285 | −0.202 | −0.197 | 0.015 | **0.768** |

Note:
Bold characters mean the significant correlation at the $p < 0.05$.

carbohydrates (*Brown et al., 1997*; *Tibbetts, Milley & Lall, 2015*). The degradation process can produce many secondary metabolites and final products, which probably change the acid-base environment, that is, organic acids (*Steinberg, 2003*). Meanwhile, the strong positive correlation between DIC and pH ($r = 0.581 – 0.817$, $p < 0.05$) as well as high pH values (>8.0) after 26 days of incubation indicated that the growth of microbes utilized DOC to generate $CO_2$ (DIC) in groups SM, SH, and M (*Shao et al., 2013*; *Yan et al., 2017*), causing the water column to become alkaline. In summary, cyanobacteria decomposition can affect redox (anaerobic dominated) and the acid-base environment in the water column.

## Influence of cyanobacteria biomass on the release of DC in overlying water

Variations in TDC, DOC, DIC, and the DOC/TDC ratio during incubation in overlying water are shown in Fig. 5. Obviously, the more cyanobacteria biomass was added, the higher TDC and DOC concentrations were detected. The box-plots also showed that TDC and DOC values in group SH added the high cyanobacteria biomass were much higher than in group S without added cyanobacteria biomass and in group SM added the medium cyanobacteria biomass (Figs. 6A and 6B). During cyanobacteria decomposition, various organic carbon compounds found in cyanobacteria are released into the water column as DOC and POC, while different carbon fractions and other elements migrate and transform (*Brocke et al., 2015*; *Chen et al., 2016*). In this study, different concentrations of cyanobacteria biomass released different organic carbon compound content into the overlying water under the same physicochemical and biological conditions. In the initial 16 days, the cyanobacteria layer was greater than zero cm in thickness (Fig. 3), DOC and DTC concentrations gradually increased (Figs. 5A and 5B), and there was little changes in DIC concentrations in groups SM and SH (Fig. 5E). This indicated that the release rate of DOC was higher than the conversion rate in overlying water. The more cyanobacterial biomass was added, the higher the peak value of DOC concentrations (S, 153.9 mg $L^{-1}$; SM, 835.1 mg $L^{-1}$, and SH, 1823.7 mg $L^{-1}$). After 16 days, compared with small changes in DIC and DOC concentrations in group S, the obvious decrease in TDC and DOC content and the DOC/TDC ratio, along with the rapid rise in DIC concentrations in groups SM and SH demonstrated that DOC transformed into DIC in overlying water. This was similar to the

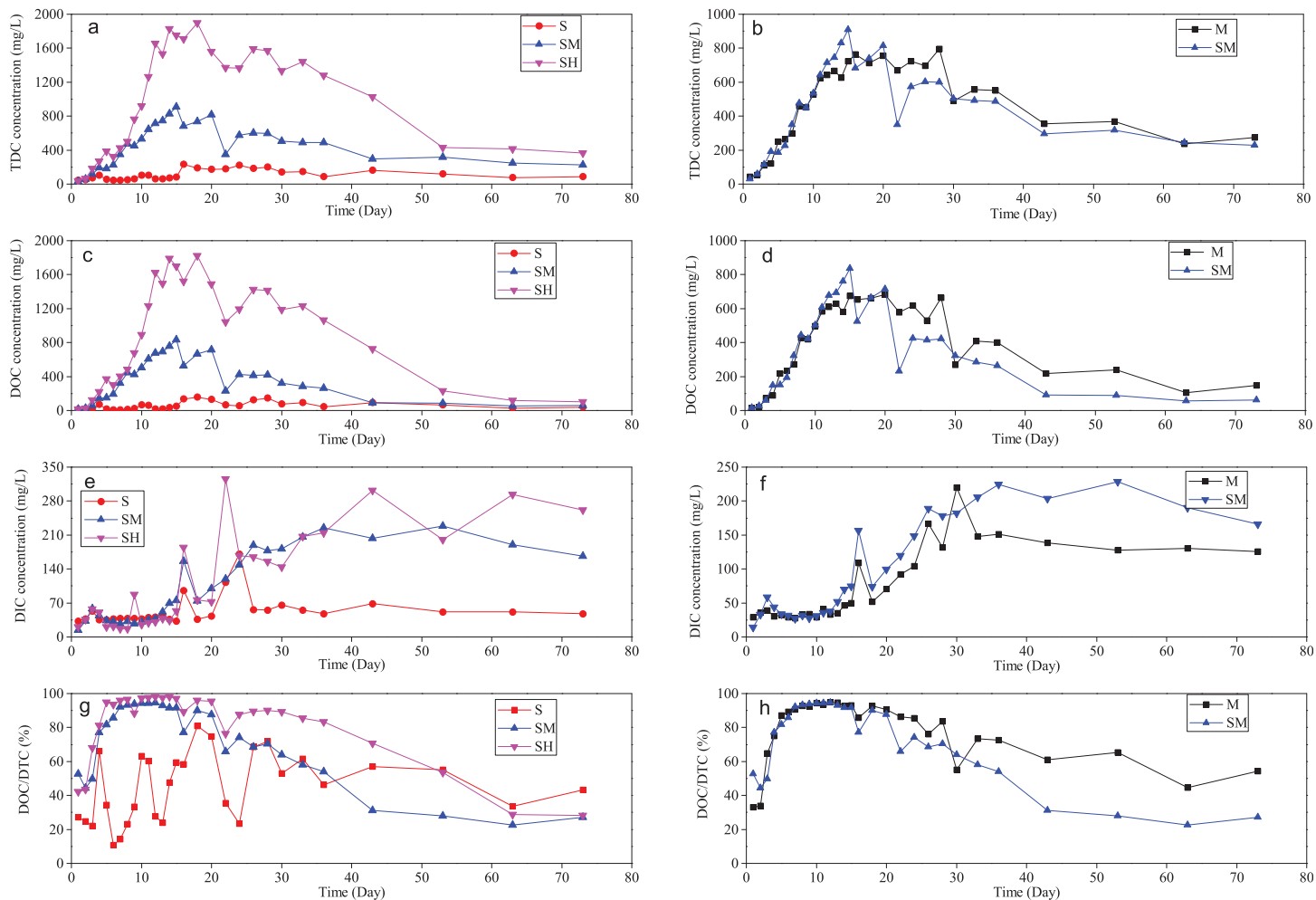

**Figure 5 Variations of TDC, DOC, DIC, and DOC/TDC ratios during incubation in overlying water of each treatment including the influence of cyanobacteria biomass and absence or presence of sediments.** TDC (A, B), DOC (C, D), DIC (E, F), and DOC/TDC ratios (G, H). The influence of added cyanobacteria biomass (A, C, E, G). The influence for absence or presence of sediments (B, D, F, H).

aquatic macrophytes decomposition (*Sobek et al., 2006*; *De Almeida Assuncao et al., 2016*), probably involving the microbial mineralization of dissolved humic substances (*Steinberg, 2003*; *Marie et al., 2015*). In addition, the treatment added the medium cyanobacteria biomass (group SM) presented the lower DOC/TDC ratio than the high cyanobacteria biomass (group SH) associated with the lower mineralization from DOC to DIC. This is probably because the decomposition of high cyanobacteria biomass consumed more oxygen and disturbed a recovering redox state. Cyanobacteria biomass variously affected the release of organic carbon and oxidative transformation from DOC to DIC.

In the microcosm with more biomass added, the pH values in overlying water were lower (Fig. 4C), probably due to more organic acids being produced by more cyanobacteria biomass during the decomposition processes. Generally, the more cyanobacteria biomass added, the higher DIC content detected (Fig. 4E). This trend indicated that the decomposition of more cyanobacteria biomass can release more carbon

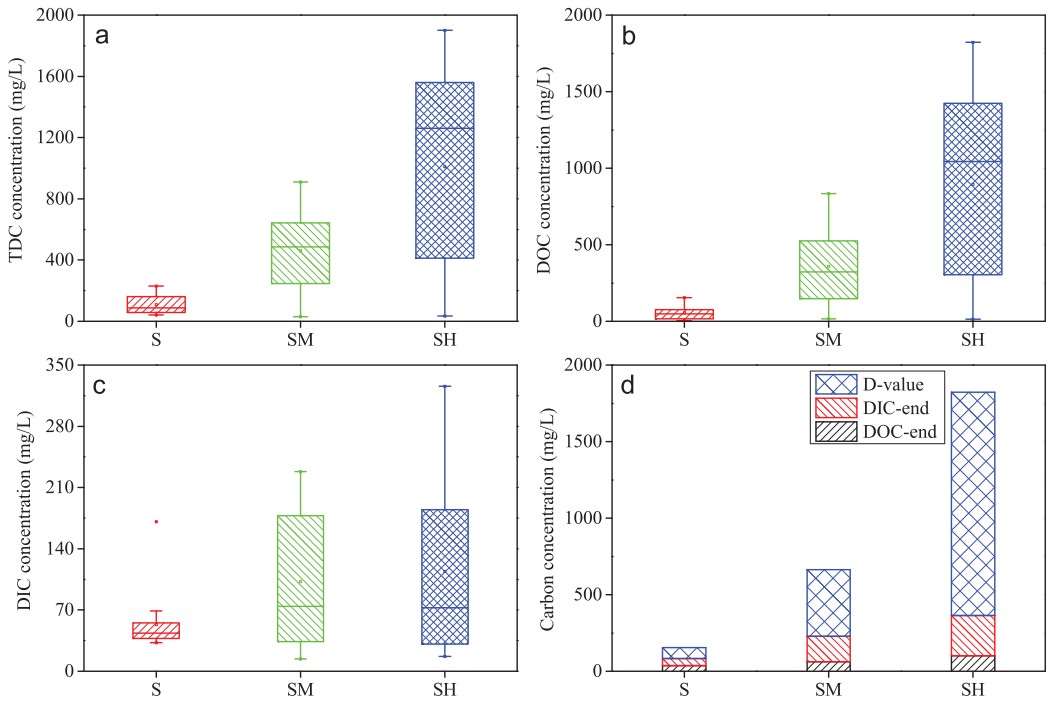

**Figure 6 Statistical box-plots of TDC, DOC, and DIC and comparison of *D*-values in treatments added different cyanobacterial biomasses.** TDC (A), DOC (B), and DIC (C) and comparison of *D*-values (D). *D*-value means the max TDC concentration in incubation subtracts the sum of DOC and DIC values in final stage in each treatment.

substrates and nutrients supporting microorganism growth in the water column, corresponding to the differences of TN, TP, $NH_4^+$-N, $NO_3^-$-N, and DOC between open water and accumulated cyanobacteria area of Taihu Lake (Table 1). The growth of microbes can promote the microbial mineralization of more DOC and cyanobacteria-derived POM into DIC (*Shao et al., 2013*; *Yan et al., 2017*). According to the carbon budget calculation approach in previous studies (*Sobek et al., 2006*; *Cremona et al., 2014*), the differences (*D*-values) between the max TDC concentration during the incubation time and the sum of DOC and DIC values in the final stage can clarify the amount of migration and transformation of TDC into sediment or air, partly based on the presence of odorous substances (*Smith, Boyer & Zimba, 2008*; *Yu et al., 2016*; *Liu et al., 2017*). *D*-values comparison between incubation groups is shown in Fig. 6D. Treatments with more cyanobacteria biomass had higher *D*-values. This confirmed that the amount of migration or transformation of DC was related to the amount of added cyanobacteria biomass. In summary, the added cyanobacteria biomass can affect the acid-base environment in the water column, migration between water and sediment phases, and transformation processes of DC involving the microbial degradation and mineralization during cyanobacteria decomposition.

## Influence of sediment on the release of DC in overlying water

Comparisons of DC release concentrations between presence and absence of sediment during the incubation time are shown in Figs. 5B, 5D, 5F and 5H. In the first 16 days,

TDC, DOC, and DIC contents with sediments (group SM) were much closer to the treatments without sediments (group M). After that time, however, along with the disappearance of cyanobacteria at the top of overlying water, the releasing intensity of organic carbon became weak. In overlying water, higher values of TDC, DOC, and DOC/TDC ratios in group M but higher DIC values in group SM illustrated a faster DOC degradation rate in group SM. Meanwhile, the thickness of the sedimentary detritus layer in group M was much higher than in group SM. This indicated a faster decomposition of detritus in group SM. These discrepancies were probably related to the addition of sediments. In our microcosm, the abiotic (photochemical) mineralization made a limited contribution to DOC transformation into DIC because of the protection from light for the entire sides of the plexi-glass tubes. Instead, the microbial mineralization might play a major role in these processes (*Obernosterer & Benner, 2004*). In group SM, the added sediments containing various bacterial communities elevated the microbial abundance and diversity of community structure (*Wang & Chen, 2008*; *Woszczyk, Bechtel & Cieslinski, 2011*; *Shi et al., 2017*), compared with group M only added lake water. Moreover, the added sediments can provide the nutrients to support microbial growth in the water column (*De Vittor et al., 2016*). By contrast with group M, the more abundant microorganisms in quantity and type involved the mineralization from DOC into DIC in group SM added sediment. Additionally, DOC molecules in overlying water could migrate to the sediments because of the concentration gradient of DOC at the sediment-water interface. Therefore, the sediments containing the abundant microbial populations and the high-level nutrients probably promoted the decomposition of cyanobacteria-derived detritus by microorganisms as well as in overlying water the microbial mineralization from DOC to DIC.

## Sediment-water interface DC processes during cyanobacteria decomposition

Dissolved carbon variations at different depths and the diffusive flux of DOC at the sediment-water interface during incubation in treatments with sediment (groups S, SM, and SH) are shown in Fig. 7. The distribution of DC with sediment depth has been shown to illustrate the migration and transformation of organic carbon at the sediment-water interface (*Burdige et al., 1999*; *Vreca, 2003*; *Chen et al., 2017*). In group S, TDC, and DIC concentrations had clear differences at sampling depth but no obvious stratification of DOC concentrations. Therefore, the DIC concentration distribution determined the clear stratification of TDC concentrations, when no cyanobacteria biomass decomposed or organic carbons were released in the water-sediment system. The DIC concentration gradient indicated that DIC migrated from pore water to overlying water at the sediment-water interface (*Wu et al., 1997*; *Vreca, 2003*). Compared to group S without added cyanobacteria biomass, the DOC concentration gradually decreased with sediment depth after adding cyanobacteria biomass in groups SM and SH (Figs. 7D–7F). This was because cyanobacteria decomposition released DOC into the overlying water and increased the DOC concentrations. Contrary to gradients found in previous studies (*Alperin et al., 1999*; *De Vittor et al., 2016*), these DOC gradients in groups SM and SH

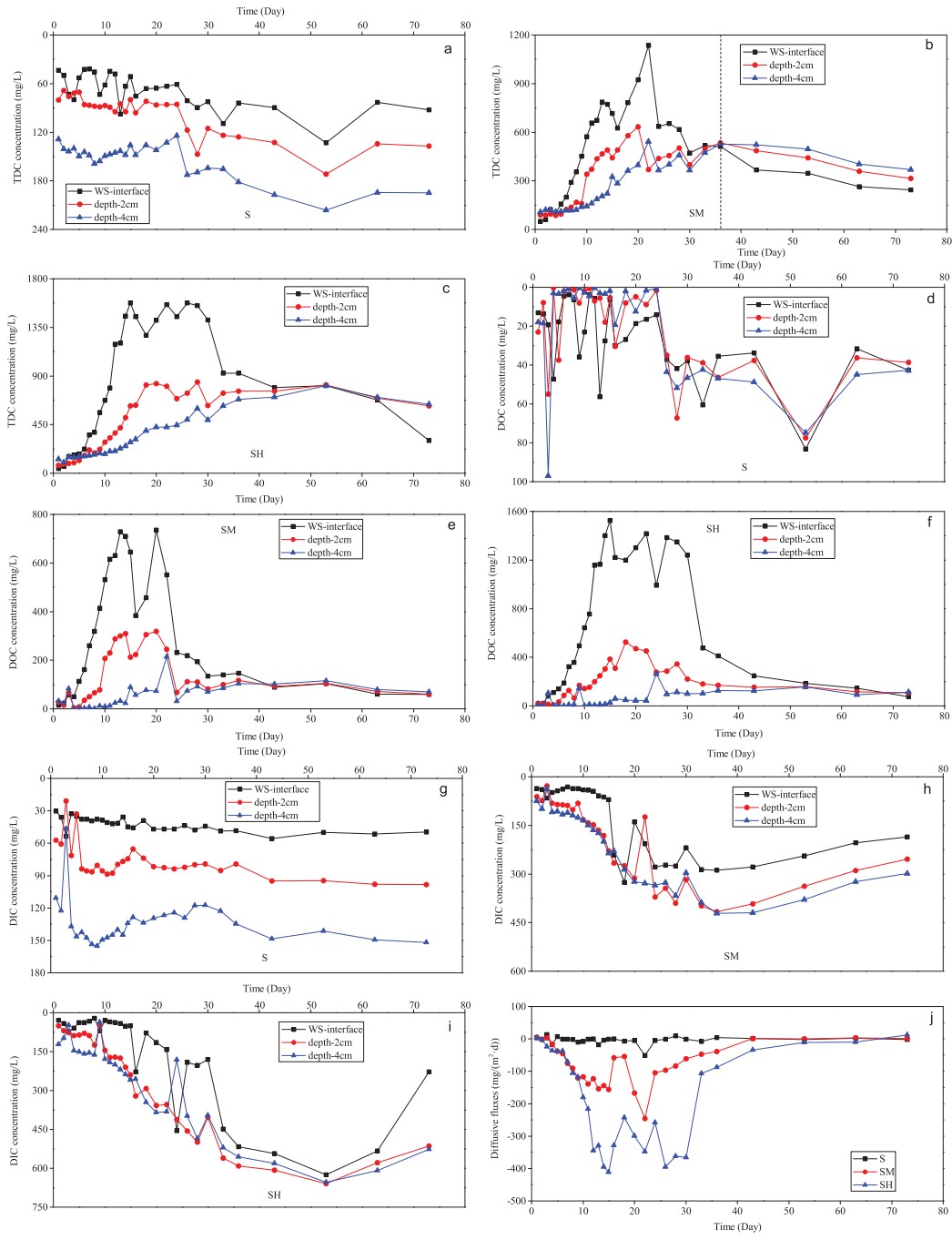

**Figure 7 Variations of TDC, DIC, DOC, and diffusive fluxes of DOC at the sediment-water interface with incubation time in each treatment added different cyanobacteria biomass.** TDC (A–C), DIC (E–G), DOC (G–I), and diffusive fluxes of DOC (J).

revealed that molecules of dissolved organic compounds in overlying water diffused into pore water at the water-sediment interface. After 43 days of incubation, the groups SM and SH showed that the DOC contents at the bottom water was close to values at depths of two and four cm due to the diffusion. The more biomass added, the higher DOC concentrations were detected at each depth, corresponding to the mass balance on

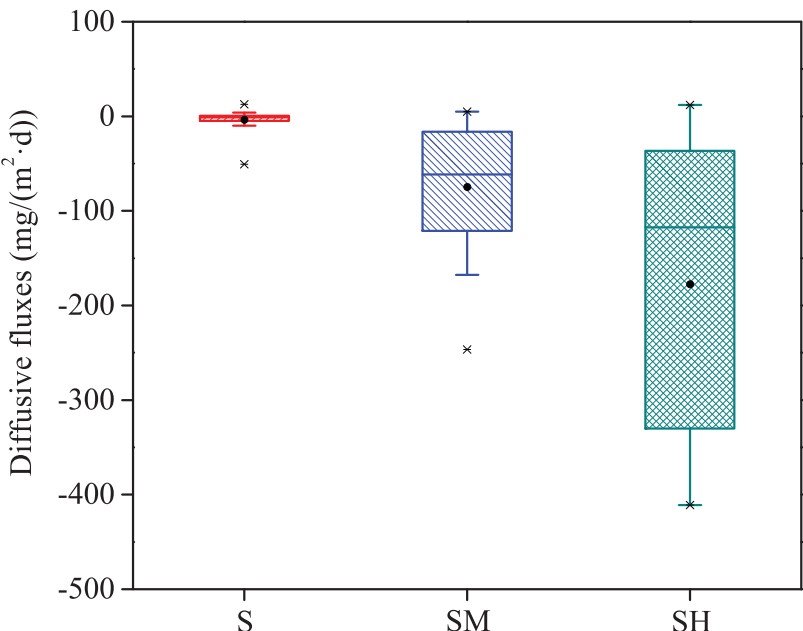

**Figure 8 Statistical box-plot of the diffusive fluxes of DOC at the sediment-water interface in each experimental group.**

carbon budget (*Hu et al., 2011*; *Cremona et al., 2014*). Compared with group S, the groups SM and SH had small differences in DOC concentration stratification with sediment depth, while DIC concentrations first increased with depth and then slightly decreased (Figs. 7G–7I). In addition, most DIC values in groups SM and SH were higher than in group S. The small differences and higher DIC values indicated that DOC in overlying water of groups SM and SH diffused to pore water and transformed into DIC. This transformation may have resulted from the microbial mineralization of DOC in sediment (*Lojen et al., 2004*; *Fahrner et al., 2008*; *Clayer, Gobeil & Tessier, 2016*). After 40 days, in the treatment added the medium biomass (SM) and after 55 days in the treatment added the high biomass (SH), the DOC contents were lower and DIC contents were slightly falling at each depth, compared with the early incubation period. These changes and the associated *D*-values (Fig. 6D) indicated that gaseous carbons might be generated in the water-sediment system and then released into air, corresponding to in-situ measurements and the carbon model of previous studies (*Kuntz et al., 2015*; *Liu et al., 2017*; *Yu et al., 2016*). Additionally, a certain amount of DIC in groups SM and SH probably migrated to the overlying water at the water-sediment interface, which prompted the increase of DIC concentrations in overlying water (Fig. 5E).

The values of DOC flux in groups SM and SH firstly increased and then decreased with incubation time, while group S presented the less DOC flux and little change (Fig. 7J). In general, the diffusive fluxes of DOC at the water-sediment interface increased with the cyanobacteria biomass added in the same time period and displayed a broader range in group SH than groups S and SM (Fig. 8), corresponding to the vertical variation and the statistical distribution of DOC concentration during cyanobacteria decomposition (Figs. 6B and 7D–7F). The diffusive fluxes of DOC reached the maximum

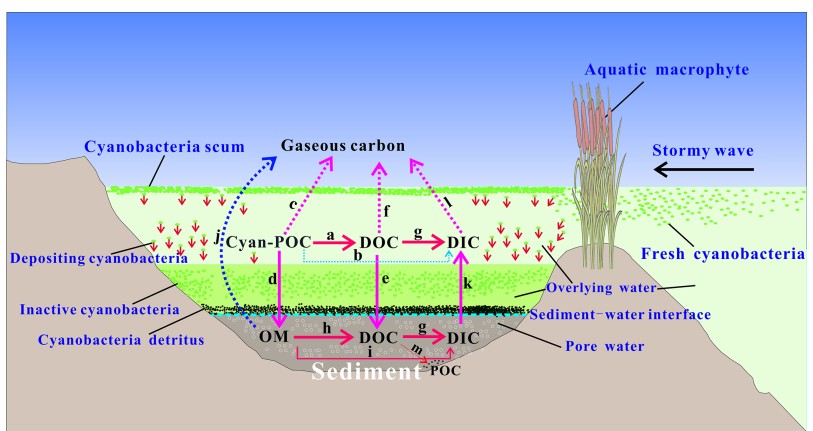

**Figure 9 Migration and transformation of DC generated by accumulated cyanobacteria in aquatic plant-belts and trenches of Taihu Lake, China.** Solid arrow and dotted arrow represent the implications of this study and the previous studies, respectively. Cyanobacteria-derived POC was biologically degraded into DOC, and may result in the decrease of pH values and anoxic state in lake water (a). A little Cyanobacteria-derived POC decomposed into DOC (b). Cyanobacteria-derived POC were transformed into gaseous carbon (*Smith, Boyer & Zimba, 2008*; *Liu et al., 2017*; *Huang et al., 2018*) (c). Cyanobacteria-derived POC deposited into organic matter at the top of sediment and contribute to humification and carbon burial (*Hu et al., 2011*; *Kuntz et al., 2015*; *Marie et al., 2015*) (d). DOC diffused from overlying water to pore water (e). A little DOC volatilized into Atmosphere (*Yu et al., 2016*; *Liu et al., 2017*; *Huang et al., 2018*) (f). DOC mineralization (g). Organic matter decomposition (h). Organic matter mineralization (i). Organic matter methanogenesis produced $CO_2$ and $CH_4$ (*Clayer, Gobeil & Tessier, 2016*; *Yan et al., 2017*) (j). DIC diffusion at the water-sediment interface causing the increase of pH value (k). $CO_2$ diffusion at the water-air interface (*Yan et al., 2017*) (l). Some DOC may be also assimilated by bacteria thus transformed to POC (*Norrman et al., 1995*; *Khodse & Bhosle, 2011*; *Yang et al., 2016a*) (m).

value of 411.01 mg m$^{-2}\cdot$d$^{-1}$ in the treatment added the high cyanobacteria biomass (group SH), which was much higher than in the normal water column (*Chen et al., 2017*). Moreover, the diffusive flux of DOC at the water-sediment interface was negative, indicating that diffusion of DOC from overlying water toward sediment was occurring. In the latter 16 days, the diffusive flux of DOC gradually declined along the decrease of DOC concentrations in groups SM and SH. Therefore, the decomposition of cyanobacteria in lakes can affect the migration process and the diffusive flux of DOC at the sediment-water interface.

## Implications for the effects of cyanobacteria decomposition on carbon cycling in lakes

To clarify the feedback of cyanobacteria-derived particulate organic carbon associated with the migration and transformation of DC during accumulated cyanobacteria decomposition in shallow eutrophic lakes, a conceptual diagram is presented according to the DC study of this microcosm system and the previous studies involving the developed model of carbon cycling (*Hu et al., 2011*), bacterial community dynamics (*Shao et al., 2013*; *Shi et al., 2017*) and the releasing mechanism of gaseous carbon (*Yu et al., 2016*; *Yan et al., 2017*) during cyanobacterial decomposition (Fig. 9). In eutrophic lakes, the summer monsoon drives a large amount of cyanobacteria blooms to

drift near lakeshore. A certain biomass of cyanobacteria gradually accumulates in lakeshore and spreads with difficulty due to the obstruction of macrophyte-belts or less influence of stormy waves in grooves, thus forming abundant scum with POC. The microcosm results and previous studies indicated that the decline and decomposition of these cyanobacteria scum will change the physicochemical environment (DO, pH, and Eh) and the microbial environment (bacterial community) of lake water (*Yan et al., 2017*; *Shi et al., 2017*). Meanwhile, the releasing DOC during cyanobacterial decomposition along with the migration and transformation of DC in water-sediment systems, alter the carbon balance at the water-sediment interface and carbon budget of overlying water and pore water in lake ecosystems. These variations for physicochemical and biophysical environment as well as DOC concentration have a profound effect on carbon cycling in lakes. Therefore, the dynamic state of DC during accumulated cyanobacteria decomposition is an important ecological indicator of biogeochemical processes involving the carbon cycling in shallow eutrophic lakes. These biogeochemical processes mainly include the decomposition of cyanobacteria-derived POC into DOC, transformation of cyanobacteria-derived POC into gaseous carbon, the settlement and burial of cyanobacteria-derived POC into sediment, DOC diffusion at the water-sediment interface, the mineralization and methanogenesis of DOC into DIC, $CO_2$, and $CH_4$, the decomposition of sediments organic matter into DOC.

Moreover, based on visual assessments in the field, the accumulated biomasses of cyanobacteria present the spatial heterogeneity in lake ecosystems due to the differences of lakeshore topographies, lacustrine currents, meteorological factors, and ecological types in accumulation areas of cyanobacteria blooms. For example, cyanobacteria blooms in eutrophic lakes are not easy to gather near estuaries but tend to be trapped in macrophyte-belts of lakeshore (*Liu et al., 2017*; *Yan et al., 2017*; *Huang et al., 2018*). The release characteristics of DOC in microcosm added diverse algae biomasses implied that the higher accumulated biomass in lakes can release the more abundant DOC into water column, having the greater potential to change the carbon budget and balance between overlying water and pore water. Meanwhile, the decomposition of diverse cyanobacteria biomasses can cause the differences of physicochemical and biophysical environment and nutrient release (Table 1) among these accumulated cyanobacteria areas, which will have various impacts on the migration and transformation of DC in spatial scale of eutrophic lakes. Therefore, the qualitative and quantitative dynamic of carbon cycling in eutrophic lakes were significantly affected by the cyanobacteria biomass.

## CONCLUSIONS

This study showed that the decomposition of cyanobacteria bloom led to the release of DOC into water column and thus changed the carbon cycling in shallow eutrophic lakes. When the variation range of temperature was not large, the biomass of accumulated cyanobacteria as well as the property of lacustrine sediments can influence the decomposition of cyanobacteria-derived POM to DOC, the mineralization of DOC to DIC, and diffusion processes of DOC and DIC at the water-sediment interface in shallow

eutrophic lakes. Furthermore, associated with previous studies about the biogeochemical mechanism of gaseous carbon and solid carbon in lake ecosystems, our study indicated that the dynamic state of DC during accumulated cyanobacteria decomposition is an important ecological indicator of biogeochemical processes in the lacustrine carbon cycling, including the decomposition of cyanobacteria-POC into DOC, the transformation of cyanobacteria-POC into gaseous carbon, the settlement and burial of cyanobacteria-POC into sediment, DOC diffusion at the water-sediment interface, the mineralization and methanogenesis of DOC into DIC, $CO_2$, and $CH_4$, the decomposition of sedimentary organic matter into DOC. In the decomposition process of cyanobacteria blooms, more research effort toward detailed analysis of the biogeochemical fate of cyanobacteria-derived organic carbon in aqueous phase, gas phase and solid phase are needed to obtain a better understanding the influence of cyanobacteria bloom and eutrophication on the carbon cycling of shallow lakes.

## ACKNOWLEDGEMENTS

We are particularly grateful to Mr. Xianshen Liu and Mr. Shuang Chen for their kind assistance in samples collecting and laboratory analyses.

### Funding

This project was supported by the National Natural Science Foundation of China (41573061 and 41703105), Provincial Natural Science Research Foundation of Colleges and Universities in Jiangsu Province (17KJB170009), Research Projects of Water Environment Comprehensive Management in Taihu Lake of Jiangsu Province (TH2014402), Provincial Natural Science Research Foundation of Colleges and Universities in Anhui Province (KJ2017A445), and Opening Foundation of Collaborative Innovation Center of Regional development in Suzhou City, Anhui Province (2016szxt02). The funders had no role in study design, data collection and analysis, decision to publish, or preparation of the manuscript.

### Grant Disclosures

The following grant information was disclosed by the authors:
National Natural Science Foundation of China: 41573061 and 41703105.
Provincial Natural Science Research Foundation of Colleges and Universities in Jiangsu Province: 17KJB170009.
Research Projects of Water Environment Comprehensive Management in Taihu Lake of Jiangsu Province: TH2014402.
Provincial Natural Science Research Foundation of Colleges and Universities in Anhui Province: KJ2017A445.
Opening Foundation of Collaborative Innovation Center of Regional development in Suzhou City, Anhui Province: 2016szxt02.
## Competing Interests

The authors declare that they have no competing interests.

## Author Contributions

- Zhichun Li conceived and designed the experiments, performed the experiments, analyzed the data, prepared figures and/or tables, approved the final draft.
- Yanping Zhao analyzed the data, authored or reviewed drafts of the paper, approved the final draft.
- Xiaoguang Xu conceived and designed the experiments, approved the final draft.
- Ruiming Han conceived and designed the experiments, approved the final draft.
- Mingyue Wang performed the experiments, approved the final draft.
- Guoxiang Wang conceived and designed the experiments, contributed reagents/materials/analysis tools, approved the final draft.

## Data Availability

Raw data are provided in the Supplemental Files.

## Supplemental Information

Supplemental information for this article can be found online at http://dx.doi.org/10.7717/peerj.5922#supplemental-information.

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
