# Peer review of "Migration and transformation of dissolved carbon during accumulated cyanobacteria decomposition in shallow eutrophic lakes: a simulated microcosm study"

_PeerJ, doi:10.7717/peerj.5922_

## Round 0.1 · original submission · Major Revisions

I received the comments from three experts. It is suggested this study is interesting in providing information regarding the carbon cycling and Cyanobacerial bloom, which is a big problem for decades in Taihu Lake. However, they also raised substantial concerns in writing, title, material & methods, discussion, and also explanations. Please find more details for their comments as below. I am happy to know your replies to these concerns.

Reviewer 1 ·

Basic reporting

no comments

Experimental design

well designed. but still need some improvement

Validity of the findings

Some conclusions are not well supported by the data in the study

Additional comments

The manuscript written by Li et al. detailed changes in dissolved organic and inorganic carbon in both overlying water and pore water was monitored and diffusive fluxes of DOC at the sediment-water interface was calculated. It is significant in understanding carbon cycling in water- sediment interface after settlement of cyanobacterial biomass. However, some revisions should be done to improve the manuscript.
1. Some conclusions are obvious, for example, more cyanobacteria more DOC. Suggest to analyze contributions of per amount of cyanobacterial biomass to DOC and DIC and make a conclusion.
2. Suggest monitoring POC concentration which would be vital for description of carbon cycling.
3. The title should be corrected. Actually, the research was conducted in microcosm system in laboratory, and just using the sediment and cyanobacteria collected from Lake Taihu. The system is much simper than the situation in situ. So “in shallow eutrophic lakes” in the title should be changed to “in simulated microcosm system”, or something like that.
4. Environmental conditions in microcosm were different from lake environments. Discuss how the results from the microcosm were applied to interpret the effects of cyanobacterial biomass decomposition on carbon cycling in lakes.

Some minor comments
Line 22, Provide the biomass for the increased TDC of 1900.0mg/L?
Line 28, what did the low biomass referred to, was it general?
Line 29-30, What is the reason for this transportation
Line 33-34, “with the maximum value of 411.01 mg/(m2·d).” what is the cyanobacterial biomass corresponding to this value?
Line 35, “presented a faster degradation of DOC” should be “presented a faster degradation to DOC”?
Line 37, “biological structures of lacustrine sediments” is not investigated in the paper, should limited to addition of cyanobacteria.
Line 41, both gaseous carbon and cyanobacteria-derived POC were not detected, so the conclusion was not well supported by the data provide in the study.
Line 130-131, the weight is wet or dry? It is better to indicated by biomass.
Line 134-135, the biomass of cyanobacteria was 5 cm-10 cm in accordance with that in field, was the length of plexi-glass tube coincides with the height of water column?
Line 186, decomposition process of cyanobacteria began after 10 days? From the data of figure 5, DOC began to increase much earlier than 10 days. So the conclusion was arbitrary.
Line 202 “with added cyanobacteria biomass” refers to all tubes with addition of various amounts of cyanobacteria or in each tube
Line 216, what are the later stages of incubation? Please give the incubation days
Line 262, it is reasonable that more cyanobacteria lead to more carbon involved in the cycling, but cycling model is similar, the study did not clearly illustrate what is the difference of migration and transformation processes of DCs among additions of different cyanobacterial biomass.
Line 351-354 the conclusion of methanogenesis, transformation cyanobacteria-POC into volatile organic carbon was not directly supported by the data in this study.
In figure 9, the diagram of carbon transformations is well known. Suggest to quantity the amount of transformation between carbon based on the result of this study. In addition, some DOC may be also assimilated by bacteria thus transformed to POC.

Reviewer 2 ·

Basic reporting

This study is to test the migration and transformation of dissolved carbon during accumulated cyanobacteria decomposition in shallow eutrophic lakes. The author find that the decopmosition of the accumulated cyanobacteria can change the carbon cycling in eutrophic lakes and will possible induce the cyanobacteria related water pollution. The paper is well designed and discussed detailly. However, there are still some questions problems that should be clearly stated before acceptence.

Experimental design

Line 111-113. The setence should be writed with caution.
Line 116-117. This might not the main reason which responsible for the formation of black water bloom.
line 130-134. The biomass of cyanobacteria is wet or dry? This is very important parameter in the experiment.

Validity of the findings

The abstract should be re-wrote and it seem very complex in its present state.Please make concise of the manuscript.

Additional comments

This paper descrice the dynamic carbon in wate and sediment during the decopmsition of cyanobacteria and the results were useful to the understanding of C cycling in eutrophic lakes. However, further study should relate these process with microbia activity.

Reviewer 3 ·

Basic reporting

no comment

Experimental design

no comment

Validity of the findings

no comment

Additional comments

Comments to #28549, “Migration and transformation of dissolved carbon during accumulated cyanobacteria decomposition in shallow eutrophic lakes”.

Cyanobacerial bloom has been a big problem in shallow eutrophic lakes for decades, which plays a significant role in carbon cycles in these lakes. However, the chemical and biological mechanisms involved in cyanobacteria affecting carbon cycling in shallow lakes remains largely unknown. Therefore, this manuscript “Migration and transformation of dissolved carbon during accumulated cyanobacteria decomposition in shallow eutrophic lakes” provide more information concerning this issue. This manuscript can be published after some revisions.

Major comments:
1) Abstract
Firstly, the abstract are too long and not well written. Some information are not clearly stated in the abstract.
Is it an in-site microcosm or incubation experiment in the lab (I found the answer in the end of the Instruction)? How long did the experiment go on? It was mentioned “In the early incubation” and “latter incubation”, what are the exact days? For example, L14-15 and L19-20 are repeated to the results and should be deleted.
I recommend the author re-write the abstract and make it clear and brief.

2) Materials and Methods
The 1st section of Results and Discussion is about “Visual changes and physicochemical environment associated with DC release”. However, in M&M, nothing was mentioned about how these visual changes (Fig. 3) was measured.
Do you have the chemical and biological characteristics (e.g., pH, Temp., phosphorus, and nitrogen levels, and Fe, etc.) of the lake water when collected? These data are important to estimate the chemical and biological activities during incubation.
Even if these parameters are not analyzed in this study, I am sure there are plenty data available in previous publications, which can also give a background information of the water conditions.

3) Results and Discussion
Somehow, I feel the big weakness of this study is lack of biological explanations of the transformations among of DC, DOC, and DIC. The authors gave too much emphasis to physical “immigration”. This may due to there are only carbon data in this study.
The decomposition of cyanobacteria can release carbon substrates and nutrients (P, N, and trace metals) which support microorganism growth in the water column. The growth of microbes utilize DOC to generate CO2 (DIC), which called mineralization. Microbial mineralization of DOC to DIC plays a crucial role in C cycling in freshwaters.
Another way for providing the nutrients to support microbial growth is sediment-release. This can explain the phenomenon in this study where “sediment increased the transformation intensity from DOC to DIC (L280-280)”. Therefore, it is important to know the P and N levels during these processes. The variation of phosphorus concentration may not be easy to detected, but it is possible to measure other parameters e.g., Fe concentration. Fe is usually bound by P and organic carbon in the sediment. The changes of Fe is a good indicator for the P release from sediment.
Abiotic (photochemcial) mineralization also plays a role in this process. However, there is no light-effect information in this study. In M&M, the authors only mentioned that “the upper parts of the plexi-glass tubes were opened and their entire sides protected from light” (L137-138).
I recommend the authors can add some discussions concerning the microbial and photochemical mineralization mechanisms to explain the phenomenon observed in this study.
Other comments:
(1) L48-51, please correct this sentence.
Two ambiguities:
a) what is the definition of ”sediment overlying water”? Does it mean the whole water column or only a few centimeters above sediment?
b) only dissolved organic carbon absorbs light.
(2) L107, please give information how the data of Figure 3 was measured.
(3) L128, is this 40cm×π×4.42cm2 correct? how much water added?
(4) L142, please give full name for DO and Eh when first being used.
(5) L142-143, what is the model of water quality monitor?
(6) L148-150, please give more information how TDC and DIC are measured by using multi N/C® 3100.
(7) L169-170, please indicate the significant level and also how D-value was calculated in this study.
(8) L183-184, there is no evidence for the EPS formation.
(9) L187, what do you mean with “in overlying water became thicker with low density”? thick and low density?
(10) L190-191, this is too farfetched. In this study neither Fe nor S concentration was measured, so one can not be sure this is related to the settle down of FeS. The settle down of FeS can be make sure by measuring the concentration of FE and/or S in the water.
(11) L206-207, correct the sentence, “The incubations were alkaline at the beginning of the experiment”.
(12) L279-281, add discussions concerning microbial and photochemical roles in this process, see main comments 3).
(13) L284, consider to change the order of Fig. 6 and 7. It seems Fig. 7 appears in front of Fig. 6 in the text.
(14) L349, delete “ecological indicator”.
(15) The legend of Figure 2 is misleading. Consider to change “in lakeshore of Taihu Lake” to “in the laboratory”.

---

## Round 0.2 · Minor Revisions

As you can see from the two reviewers, there are only minor concerns left, which are relevant to clarity. Congratulations on this interesting microcosm study and I expect to see your proper revision soon.

Reviewer 2 ·

Basic reporting

The quality of this paper has been improved a lot after revision.

Experimental design

The experimental design was reasonable and scientific feasible.

Validity of the findings

The finds is important to understand the carbon cycling in freshwater lakes.

Additional comments

All of the consieration has been answered and this paper can be accepted in its present situation.

Reviewer 3 ·

Basic reporting

no comments

Experimental design

no comments

Validity of the findings

no comments

Additional comments

Reviewer #3: I appreciate the authors’ revisions considering my comments on the original submission. The manuscript is now much better presented than the original version.
Some minor comments,
(1) In abstract, L32, maybe it is better to change 1823.7 mg/L to 1.82 g/L;
(2) L53-56, this long sentence is not correct or accurate. “DC can be found in the whole water column, not mainly found in …..”. I know the author want to emphasize the importance of overlying water and pore water, but this sentence needs to be modified.
(3) L125-126, what is measured? Please modify or delete this sentence.
(4) L153-155, this long sentence need to be simplified. It is not clearly stated. The same problem can be found in L221-225, a very long sentence, it is hard to catch the main point. I suggest the authors rewrite these sentences and replace with short and clear sentences.

---

## Round 0.3 · accepted · Accept

The algae soap, caused by Cynobacteria boom, is a serious problem in hypereutrophic lakes, such as Taihu Lake. The subsequent decomposition in aquatic environments is full of complex biogeochemical processes reluctant to be deciphered. The findings from the long-time microcosm experiment are a great step in understanding the migration and transformation of dissolved carbon associated with algae soap.

#